# Brief communication: Arctic sea ice thickness internal variability and its changes under historical and anthropogenic forcing

Guillian Van Achter[1], Leandro Ponsoni[1], François Massonnet[1], Thierry Fichefet[1], and Vincent Legat[2]

[1]Georges Lemaitre Center for Earth and Climate Research, Earth and Life Institute, Université Catholique de Louvain.
[2]Institute of Mechanics, Materials and Civil Engineering, Applied Mechanics and Mathematics, Université catholique de Louvain.

**Correspondence:** Guillian Van Achter (guillian.vanachter@uclouvain.be)

**Abstract.** We use model simulations from the CESM1-CAM5-BGC-LE dataset to characterise the Arctic sea ice thickness internal variability both spatially and temporally. These properties, and their stationarity, are investigated in three different contexts: (1) constant pre-industrial, (2) historical and (3) projected conditions. Spatial modes of variability show highly stationary patterns regardless of the forcing and mean state. A temporal analysis reveals two peaks of significant variability and despite a non-stationarity on short timescales, they remain more or less stable until the first half of the 21st century, where they start to change once summer ice-free events occur, after 2050.

## 1 Introduction

In the recent decades, Arctic sea ice has retreated and thinned significantly (Notz and Stroeve, 2016). The annual mean Arctic sea ice extent has decreased by $\sim 2 \times 10^6$ km$^2$ between 1979 and 2016 (Onarheim *et al.*, 2018). An analysis combining US Navy submarine ice draft measurements and satellite altimeter data showed that the annual mean sea ice thickness (SIT) over the Arctic Ocean at the end of the melt period decreased by 2 m between the pre-1990 submarine period (1958-1976) and the Cryosat-2 period (2011-2018) (Kwok, 2018). On long timescales (a few decades or more), retreating and thinning are projected to continue as greenhouse gas emissions are expected to rise. However, on shorter timescales (1-20 yr), internal climate variability, defined as the variability of the climate system that occurs in the absence of external forcing and caused by the system's chaotic nature, limits the predictability of climate (Deser *et al.*, 2014) and represents a major source of uncertainty for climate predictions (Deser *et al.*, 2012). In this context, greater knowledge of Arctic SIT internal variability and of its drivers are both essential to document the true evolution of the Arctic atmosphere–ice–ocean system and to predict its future changes.

The mean spatial distribution of the Arctic SIT is relatively well documented (Stroeve *et al.*, 2014). But there are some uncertainties around its interannual variability and its spatial modes of variability. Some studies (Lindsay and Zhang, 2006; Fuckar *et al.*, 2016; Labe *et al.*, 2018) already analysed the spatial distribution of Arctic sea ice variability by applying empirical orthogonal functions (EOF) (K-means cluster analysis for Fuckar *et al.*, 2016) to model-based historical SIT time series. Lindsay and Zhang (2006) reported a first mode nearly basinwide, while the second and third ones are orthogonal lateral modes accounting for 30, 18 and 15% of the variability, respectively. Fuckar *et al.* (2016) also found a nearly basinwide first mode,

with an Atlantic-Pacific dipole as the second mode. Labe *et al*. (2018) depicted an Atlantic-Pacific dipole but as the first mode. The spatial structure and amount of explained variance of those modes are sensitive whether and how the SIT time series is detrended. It is also model-dependent and influenced by the season and analysed period. The temporal sea ice volume (SIV) variability has been studied by Olonscheck and Notz (2017). These authors enlightened a remarkable similarity between the pre-industrial and historical internal variabilities of the annual Arctic SIV. They also noticed a decreased internal variability of winter and summer Arctic SIV for a future climate forced by the RCP8.5 scenario.

Apart from Olonscheck and Notz (2017), the studies cited above used data covering a few decades under historical forcing. In this work we use a long climate model control run under pre-industrial conditions from the CESM1-CAM5-BGC-LE dataset, which enables us to study only the internal variability of the Arctic SIT. We study the internal variability both temporally and spatially by applying a wavelet analysis and an EOF decomposition to the pan-Arctic SIV and gridded SIT anomaly time series, respectively. We also determine whether or not the SIV and SIT variability is stationary by analysing the model outputs under historical and future climate conditions with 30 ensemble members.

This manuscript is organised as follows. The model and its outputs are briefly described in Section 2. In Section 3, the spatial and temporal internal variability of Arctic sea ice are analysed, as well as their persistence through historical and future climate conditions. Then we explore the drivers of the main modes of internal variability. Conclusions are finally given in Section 4.

## 2 Data and methods

### 2.1 Sea ice thickness and volume datasets

We use the CESM1-CAM5-BGC-LE dataset (Kay *et al*., 2015). The Community Earth System Model Large Ensemble (CESM-LE) was designed to both disentangle model errors from internal climate variability and enable the assessment of recent past and future climate changes in the presence of internal climate variability. The CESM1(CAM5) is a CMIP5 participating-model. It consists of coupled atmosphere, ocean, land and sea ice component models. It also includes a representation of the land carbon cycle, diagnostic biogeochemistry calculations for the ocean ecosystem and a model of the atmospheric carbon dioxide cycle (Moore *et al*., 2013; Lindsay *et al*., 2014). While it is not possible to validate the data in terms of SIT and SIV variabilities due to the lack of continuous observational data, the model was well validated in terms of mean state of the ice thickness and extent, as well as regarding the recent trends in the latter. Jahn *et al*. (2016) showed good agreement between observations and CESM1(CAM5) simulations for mean Arctic sea ice thickness and extent in the early twenty-first century. Barnhart *et al*. (2016) demonstrated that CESM1(CAM5) captures the trend of declining Arctic sea ice extent over the period of satellite observations. Based on these validation studies, we consider that the CESM1-CAM5-BGC-LE time series is a fair proxy to study the variabilities of the Arctic SIT and SIV under different forcing conditions.

In this paper, we use the monthly averaged Arctic SIT and SIV provided over the 3 periods (pre-industrial, historical and future). The pre-industrial period is represented by a single 1700-yr control simulation with constant pre-industrial forcing. The ocean model was initialised from a state of rest (Danabasoglu *et al.*, 2012), while the atmosphere, land and sea ice models were initialised using previous CESM1(CAM5) simulations. This experimental design allows the assessment of internal climate variability in the absence of climate change. In practical terms, we will use the last 200 years of this simulation. The historical period has one ensemble member covering the 1850-2005 period and 30 ensemble members over 1920-2005. Also with 30 ensemble members, the future climate period (2006-2100) follows the representative concentration pathway (RCP) 8.5 scenario, corresponding to a total radiative forcing of 8.5 W/m$^2$ in 2100 relative to pre-industrial conditions (Meinshausen *et al.*, 2011). The Canadian Archipelago region was removed from the dataset since SIT reaches unrealistic values in this area.

For the variability analysis, the trend and seasonal cycle are removed from the time series (pan-Arctic SIV and gridded SIT) so that we focus on the interannual variability. Since the spatial variability analysis uses 30 ensemble members, the SIT anomaly fields are computed by removing the ensemble mean to each member. When only one ensemble member is used, as for the temporal analysis, the anomaly is calculated by excluding the individual trend (provided by a second-order polynomial fit) of each month.

## 2.2 Variability analysis

To characterise the internal variability of the Arctic sea ice, we aim at inspecting how the SIV variability evolves in time and how SIT variability is characterized in space. For addressing the temporal variability, we make use of wavelet analysis, with Morlet as wavelet mother, following the methodology proposed by Torrence and Compo (1998). The wavelet analysis has the advantage of taking into account possible non-stationarity of the time series. In this paper, we show the results for one of the historical (1850-2005) members and one of the future (2006-2100) members, although we tested the robustness of the results over the 30 ensemble members as discussed later (Section 3.1 and 3.2).

The spatial variability is analysed by computing the EOFs on the SIT anomaly time series. This decomposition reduces the large number of variables of the original data to a few variables, but without compromising much of the explained variance. Each EOF represents a mode of SIT variability that provides a simplified representation of the state of the SIT at that time along that EOF. In other words, the EOFs themselves are fixed in time but their weighting coefficients are time-varying; the associated time series (one for each mode) indicate in which state the SIT is at any time (Hannachi, 2004). The analysis is made on the gridded SIT anomaly time series for the 3 periods. For the historical and future periods, the EOFs are computed over 30 ensemble members, all appended together over time (as done by Labe *et al.*, 2018).

By applying those analyses separately over the 3 periods we aim to document the internal variability in the absence of any external forcing during the pre-industrial period. By comparing the pre-industrial results with those for the historical and future periods, we estimate the evolution of the SIT and SIV internal variability under anthropogenic forcing.

## 3 Results

### 3.1 Temporal variability

The results from the wavelet analysis are presented in Figure 1a-c, in which the wavelet power spectrum is shown as a function of time (bottom-left of each subfigure). On the wavelet power spectrum, the crosshatched area denotes the "cone of influence", in which edge effects become important, and the red lines denote the $95\%$ significance levels above a red noise background spectrum. The global wavelet spectrum is also shown (bottom-right), which is a time-integrated power of the wavelet power spectrum. The significance level of the time-integrated wavelet spectrum is indicated by the dashed curve. It refers to the power of the red noise level at the $95\%$ confidence level that increases with decreasing frequency.

The temporal variability of the Arctic SIV anomaly over the pre-industrial period is depicted in Figure 1a. The time-integrated power spectrum (bottom-right) shows 2 peaks of significant variability. The first peak corresponds to a period centered on 8 years but spanning from 5 to 10 years. The second one corresponds to a period of 16 years spanning from 10 to 20 years. In the wavelet power spectrum, the red lines enclose regions in which the variability is significant. The two main peaks are present throughout the time span, but not always concomitantly. Depending on the time, both the 8- and 16-year periods are significant with one of them appearing stronger in the power spectrum (Figure 1a, bottom-left panel). For instance, the 8-year peak is dominant during the 1780-1810 period, the 16-year peak during the 1750-1780 period, and both peaks are dominant during the 1830-1850 period.

Over the historical period, the Arctic SIV temporal variability shows a first peak centered on 5 years and two others centered on 10 and 16 years, all with $95\%$ confidence (Figure 1b). The wavelet power spectrum shows that the 16-year period is significant throughout the entire time span, while the 8-year period loses significance around certain periods of time (e.g., around 1925). The future climate SIV wavelet analysis in Figure 1c presents a clear loss of variability after the year 2050. This loss of variability is visible in the SIV time series and is confirmed by both the wavelet power spectrum and the time-integrated power spectrum. The 2050 sudden loss of variability coincides with the ice-free summer events occurring at that time. Apart from that loss of variability, the wavelet power spectrum exhibits one band of 5-years variability during the 2015-2025 period and another band of 10-years variability during the 2025-2050 period, both bands with $95\%$ of confidence. In Figure 1c, the peaks are not significant on the time-integrated power spectrum because the respective variability is significant only over the the first 50 years as it is shown in the wavelet power spectrum (areas in red).

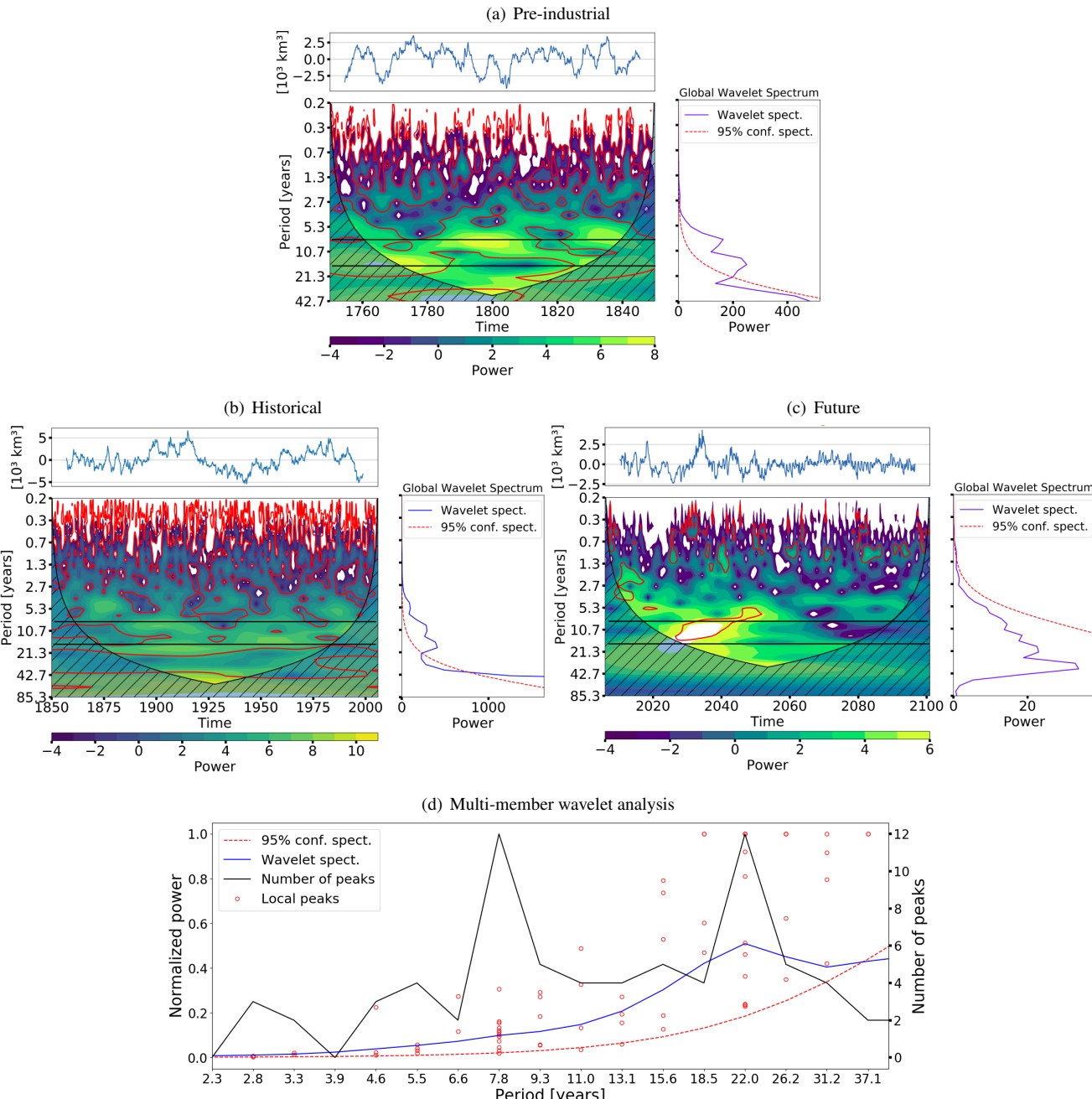

**Figure 1.** Wavelet analysis applied to the Arctic sea ice volume anomaly over the pre-industrial (200 years preceding the historical integration) (a), historical (1850-2005) (b) and future (2006-2100) (c) periods. Each subfigure (a-c) presents the sea ice volume anomaly time series (top), wavelet power spectrum (bottom-left), and time-integrated power spectrum from the wavelet analysis (bottom-right). Morlet is used as a wavelet mother. The red lines denote the 95% significance levels above a red noise background spectrum, while the crosshatched areas indicate the cone of influence where edge effects become important. White areas in the wavelet power spectrum are representing values out of the range defined by the color bar. Horizontal black lines depict the 8 and 16-year periods. Multi-members wavelet analysis (d). The red dots depict wavelet spectrum local maxima for all members. The blue and dashed red lines show the mean normalised wavelet spectrum and 95% confidence spectrum for all members, respectively. The black line represent the number of wavelet spectrum local maxima at each period.

The main characteristics of the temporal variability of the Arctic SIV under pre-industrial conditions seem to persist under anthropogenic forcing. The two major temporal peaks of variability centered on 8 and 16 years, found in the pre-industrial run, are also present during the historical period. For the first half of the 21st century, the future projections are also dominated by the two main peaks but centered at 5 and 10 years in the integrated spectrum, and with relatively weaker power compared to the pre-industrial and historical runs. Furthermore, the SIV variability seems to be non-stationary since the power is not always above the 95% significance level.

The wavelet analyses applied to the other 30 ensemble members of the historical and future simulations bring robustness to our results since, overall, each member shows a similar pattern of temporal variability. To promote such a multi-member comparison among the different spectra, we have first normalised all spectra (and the significance curve) by their respective maximum value so that the power ranges from 0 and 1. This step is required to make that the spectrum from each member has the same weight in the averaging. As shown in Figure 1d (blue line), the averaged spectrum is smoothed out across the time domain because the peaks from different spectra are not co-located exactly at the same periods. Nevertheless, it is still showing that the variability is significant over the background red noise (see dashed red line). To complement this analysis, we have counted the number of local peaks for each period and from all 30 spectra. As shown by the black line in Figure1d, there is a concentration of peaks around the 8-year and 22-year periods. This spread compared to the reference historical run is somehow expected since the internal variability between the different members is not expected to be identical, and even tends to increase with time (Blanchard-Wrigglesworth *et al.*, 2011). For members covering the 21st century, the results are close to the one-member analysis discussed above.

## 3.2  Spatial variability

The spatial variability of the Arctic SIT anomaly is depicted by the major modes of variability in Figure 2. Since the SIV exhibits a strong loss of variability around the year 2050, the future period for this spatial variability analysis spans from 2006 to 2050. For each period, the modes are sorted by percentage of variability explained. The first mode, which explains most of the variability, represents 22, 20 and 20 % of the variability for pre-industrial, historical and future climate conditions, respectively. All periods show the same pattern of SIT spatial variability for the first mode. It corresponds to a dipole between the Fram Strait area and the East Siberian Sea (Figure 2 (a,b,c)). For both the pre-industrial and historical periods, the second mode of variability is a pole centered in the East Siberian Sea, but also spreading into the Arctic Basin (Figure 2 (d,e)). It accounts for 14 and 11 % of the variability, respectively. The third mode of variability for the pre-industrial period corresponds to a dipole between the Laptev and Kara Seas, on the one hand, and the east coast of Greenland, the Chukchi sea and Beaufort Sea, on the other hand.

The first mode of SIT is stable over time and stays the dominant mode of spatial variability in all three periods. There are some disparities in percentage explained and in magnitude, which could be explained by the different lengths of the periods. As the first mode, the second mode of SIT spatial variability is persistent in the historical period. For the future climate period, the

155  second mode of SIT variability is no longer persistent. It presents a dipole of variability as the first mode, but the Pacific part of the dipole is larger and no longer located in the East Siberian Sea. The third modes of the three periods (Figure 2 (g,h,i)) exhibit all different patterns of variability and they are not considered in further analysis.

After 2050, the SIT spatial variability is impacted by the sudden decrease in SIT. EOFs computed over the 2050-2100 period
160  (not shown) exhibit the same pattern of the dipole as the first mode for the 2005-2050 period, but the area of high variability is not the same. The Atlantic part of the dipole is shifted toward the north coast of Greenland and the Pacific part of the dipole is also reduced near the coast.

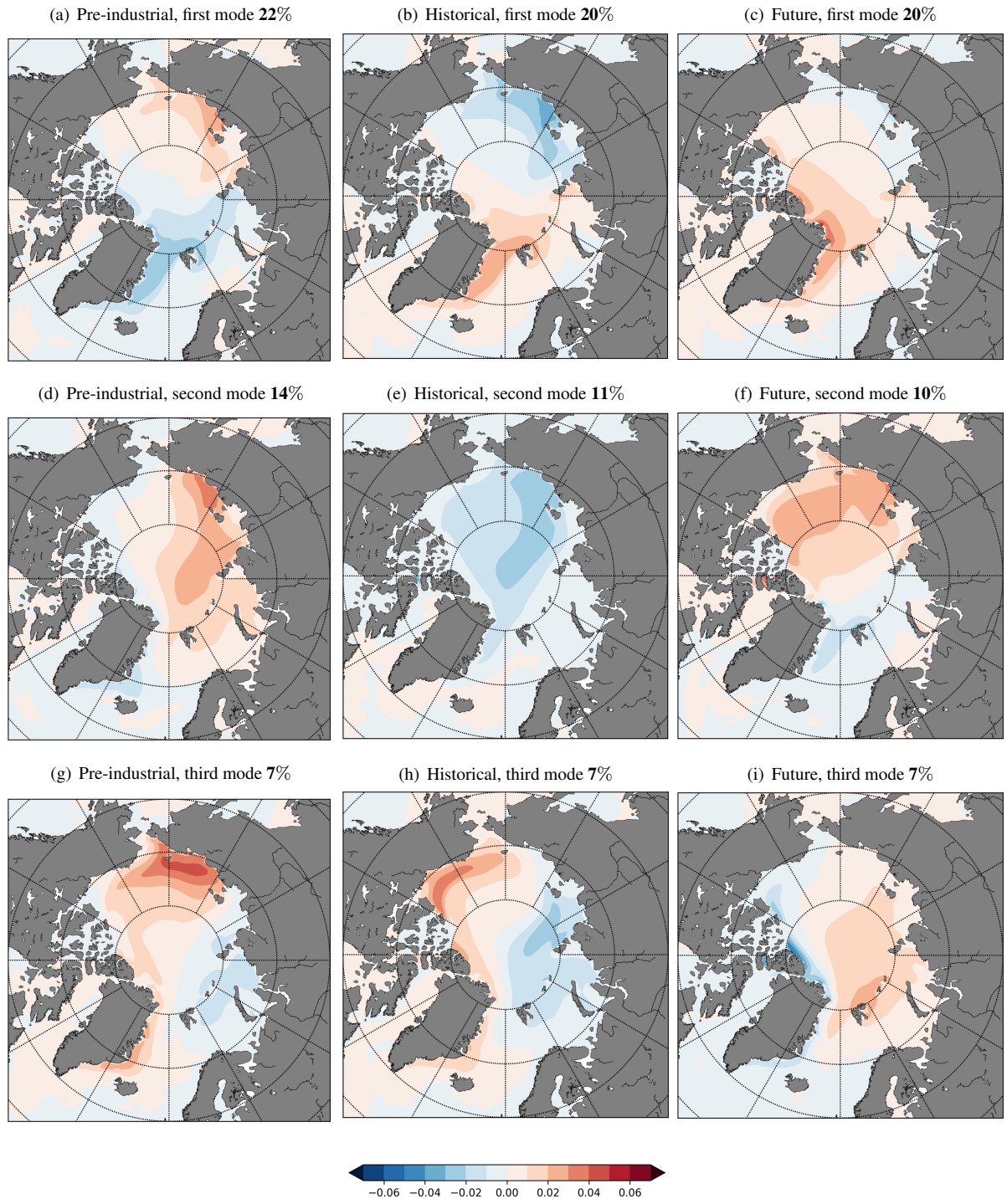

**Figure 2.** Modes of Arctic SIT spatial variability. From the left to the right, each row shows the three first EOF of Arctic SIT over the pre-industrial (200 years preceding the historical integration) (a,d,g), historical (1920-2005) (b,e,h) and future (2006-2050) (c,f,i) periods, respectively. EOFs for the historical and future periods are performed over 30 ensemble members.

### 3.3 Drivers of the major modes of SIT internal variability

By computing the temporal oscillation between phases of a certain mode of variability, we are able to characterise this mode by low and high indices. In order to find the physical drivers of the SIT modes of variability, we investigate the differences in dynamic and thermodynamic features (sea ice velocity, atmospheric surface temperature) between both phases of the modes. Figure 3a,b show the mean Arctic sea ice circulation over the pre-industrial period by compositing the low (a) and high (b) indices for the first mode of SIT variability. The sea ice drift anomaly associated with the positive and negative phases of the first SIT mode share similar features with the Arctic Oscillation: a cyclonic anomaly in the Beaufort Gyre, impacting the Transpolar Drift Stream, the Laptev Sea Gyre and the East Siberian circulation, as described by Rigor *et al*., (2002).

Furthermore, applying wavelet analysis to the associated time series of the first spatial mode of variability indicates that the main periodicity of this mode is centered on 8 years and spans from 5 to 10 years (not shown). This result is suggestive of a link between the first mode of temporal variability of the wavelet analysis and the first mode of spatial variability, and so, to the Arctic Oscillation.

We also used the associated time series of the second mode of SIT spatial variability to characterise it by low and high indices. The same analysis over the sea ice velocity is performed for the second mode. For both indices, the sea ice velocity fields are similar. We concluded that the second mode is not dynamically driven. Following Olonscheck *et al*. (2019) results, which demonstrate that the internal variability of Arctic sea ice area and concentration are primarily caused by atmospheric temperature fluctuations, we investigated the differences in mean surface air temperature anomaly over the pre-industrial period between the low and high indices for both the first and second modes of SIT variability. Two widely different states of surface air temperature are found between indices for both modes (the surface air temperature anomaly for the second mode is depicted in Figure 3c,d). It appears that the SIT variability and the surface air temperature are associated with each other.

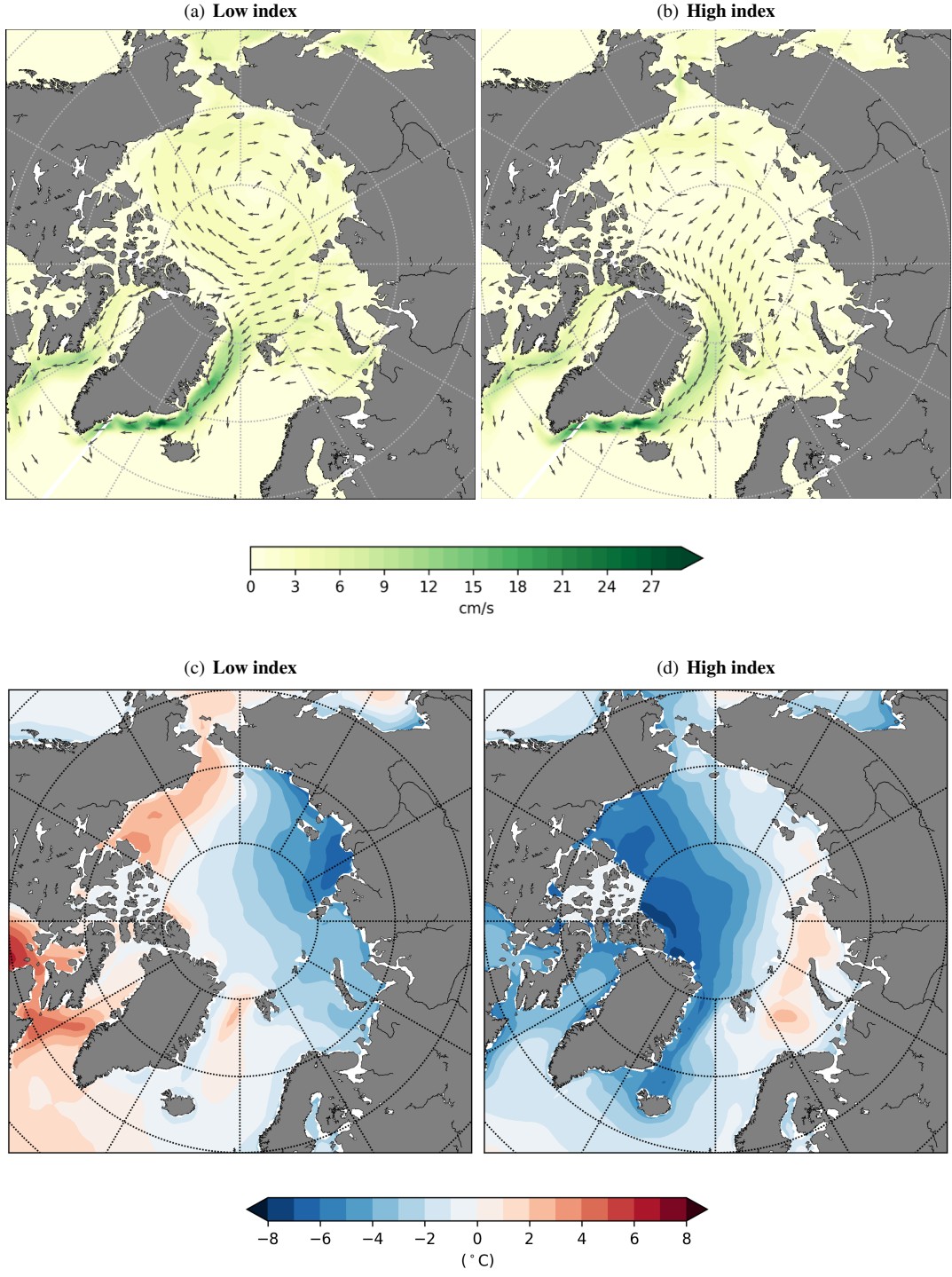

**Figure 3.** Arctic sea ice mean circulation during low (a) and high (b) indices of the first mode of SIT variability during the pre-industrial period. Arctic mean surface air temperature anomaly during low (c) and high (d) indices of the second mode of SIT variability.

 **4   Conclusions**

In this work, we have analysed the internal variability of the Arctic SIT both spatially and temporally with the CESM1-CAM5-BGC-LE dataset. We conducted wavelet analysis of the pan-Arctic SIV anomaly and EOF decomposition of the gridded SIT anomaly, both over a 200-yr control run conducted under pre-industrial conditions. Then, to assess the persistence of the SIT anomaly internal variability under anthropogenic forcing, we performed the same analyses with 30 ensemble members over the historical and future periods.

The temporal analysis of the SIV anomaly internal variability shows two peaks of significant variability. One centered on 8 years, spanning from 5 to 10 years, and another one centered on 16 years, spanning from 10 to 20 years. These two peaks of temporal variability are present in both the pre-industrial and historical periods, as well as in the first half of the 21 century. After that, a sudden loss of variability due to ice-free summer events is found. Furthermore, despite a dominant periodicity over the three periods, the SIV anomaly has been observed to be non-stationary. Indeed, the dominant periodicity of the SIV variability can be either centered on 8 or 16 years, depending on the timescale and period. Wavelet analyses over the 30 ensemble members for the post-industrial period have shown the same behaviour of temporal variability within members, except that the peaks are not always centered in 8 and 16 years but somewhere between 5-10 and 15-26 years, depending on the member.

The spatial analysis of the SIT anomaly internal variability has been applied to the 30 ensemble members and reveals two important modes of variability. The first one is a mode with opposite signs centered in the East Siberian Sea and in the Fram Strait area, accounting for 22% of the variability in the pre-industrial period. This first mode is a dynamical one, related to the Arctic Oscillation, and persists over all pre-industrial, historical and future periods. Furthermore, this first mode of spatial variability has a temporal variability of 8 years (spanning from 5 to 10 years), corresponding to the first peak of variability found in the temporal analysis. The second mode exhibits a large pole of variation centered on the East Siberian Sea going through the Arctic Basin. It represents 14% of the variability in the pre-industrial period.

The loss of sea ice in summer starting in 2050 and the strong decrease in SIV in winter during the second half of the 21st century (from 15 to $10 \times 10^3$ km$^3$) strongly modifies the variability of the ice both spatially and temporally. The main modes of spatial variability lose their significance or just disappear after 2050, and the temporal analysis shows a total disappearance of the variability at that time.

This analysis of the Arctic SIT and SIV variability bears some limits. Indeed, our results for the temporal and spatial patterns of variability are based on only one model, and despite the use of 30 ensemble members and a reasonable validation against observations, the model is not perfect. Furthermore, the spatial modes of SIT variability are robust for all the 30 ensemble members but the temporal analysis shows some dissimilarities between members. Other studies with other model outputs are

therefore needed to confirm our conclusion.

220 Finally, in the context of recent climate changes, predicting sea ice has never been so important. However, to validate and improve our predictions, observational data is crucial. In this sense, our variability analysis of internal SIV and SIT variability might help the development of an optimal sampling strategy, taking into account the selection of well-placed sampling locations for monitoring the SIT and, therefore, the pan-Arctic SIV, that are not as well documented as the sea ice extent and area (Ponsoni *et al.*, 2019).

225

*Data availability.* Data can be downloaded from the following source: https://www.earthsystemgrid.org/dataset/ucar.cgd.ccsm4.CESM_CAM5_BGC_LE.ice.proc.monthly_ave.html. The 30 ensemble members used in this study are the first 30 members (001-030).

*Code and data availability.* The wavelet analysis is performed with the Waipy module on Python. https://github.com/mabelcalim/waipy

*Competing interests.* The authors declare that they have no conflict of interest.

230 *Acknowledgements.* The work presented in this paper has received funding from the European Union's Horizon 2020 Research and Innovation programme under grant agreement no. 727862: APPLICATE project (Advanced prediction in Polar regions and beyond). We also thank the EU Horizon 2020 PRIMAVERA project, grant agreement no. 641727. François Massonnet and Leandro Ponsoni are F.R.S.-FNRS Research Associate and Post Dotoral Researcher, respectively. Guillian Van Achter is founded by PARAMOUR project which is supported by the Excellence Of Science programme (EOS), also founded by FNRS. We thank the two referees for their very helpful comments on a 235 earlier version of this manuscript. The dataset used in this study was made available by CESM Community.

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
