# Peer review of "Brief communication: Arctic sea ice thickness internal variability and its changes under historical and anthropogenic forcing"

_The Cryosphere, 2019_

## Referee Comment (RC1) · Anonymous Referee #1 · 7 Apr 2020

**Review of: "Brief Communication: Arctic sea ice thickness internal variability and its changes under historical and anthropogenic forcing", by Guillian Van Achter et al**

**General comments**

This brief communication analyses the variability of Arctic sea ice thickness in pre-industrial, historical and future climate simulations from the CESM1(CAM5) coupled model. Both temporal analysis of the timeseries of sea ice volume, and spatial analysis of the sea ice thickness are presented, and results from the two analyses are brought together in the discussion.
The main findings are that this model shows two peaks of temporal variability (8 and 16 years) in the pre-industrial simulation, which persist in the historical simulation, and until the middle of the 21$^{st}$ century. The first mode of spatial variability is a dynamic mode related to the AO, and corresponds to the 8 year peak in temporal variability. Both the spatial and temporal variability change significantly from the 2050s when the summer sea ice is lost.

In terms of the originality, scientific quality, significance and presentation quality I asses this communication as good. The application of wavelet analysis to the SIV timeseries is interesting, and it is really good to see the temporal and spatial analysis brought together in the same piece of work.

Overall, I feel the paper could benefit from
-   More clarity in the details of the analysis, and some of the explanations.
-   Some improvements to figure 1
-   A better bringing together of thoughts at the end of the conclusions.

I hope these comments will prove useful.

**Specific comments**

Lines 46-50: While the model has been well validated for the mean state of ice thickness and extent, and for the declining in ice extent, it is of course not possible to validate the variability in the ice thickness/volume, and so I think the statement that it *can be* assumed that the modelled timeseries is an adequate proxy is perhaps too strong - the assumption is a caveat of the work.

Lines 53-54: Was the analysis done using just one of the historical and future climate ensemble members? I assume so, but it would be good to clarify this. As an aside, it would be interesting to know how robust this analysis is if it is applied to different members of the ensemble.

Lines 59-64: I found this paragraph a little confusing. The removal of the trend and seasonal cycle from monthly data for the SIV timeseries is clear, but I was less sure exactly what was done for the SIT fields before the EOF analysis. Later the analysis of Lindsay and Zhang is mentioned – they used annual mean data, so it would be good to clarify exactly what was done.

Paragraph beginning at line 67: I found this paragraph difficult to follow on first reading. I think it would help the reader to start the second sentence in a way that makes it clearer that the discussion will initially focus on the temporal analysis (the start of the following paragraph is much clearer in this regard).

Figure 1: I have a number of suggestions that would make this figure easier to follow:

- The discussion in the text refers to the time periods in years, whereas the scale in the figure is in months – it would be easier to follow if the scale was also in years.
- Perhaps the lines marking the areas of significant variability could be a colour not used in the scale, so that they stand out more. This is especially needed in 1c, where there is more yellow on the plot itself.
- Maybe the 8 and 16 year periods could be marked by horizontal lines (on the panes representing the wavelet power spectrum).
- I'm not sure why the Fourier spectrum is included with the time-integrated power spectrum, as I don't think it is mentioned in the text.

In addition, the meaning of the hatched area is not explained anywhere.

Lines 110-111: It looks like the peaks discussed here are not significant? In the discussion of Fig 1a, the 42 year peak is not discussed because it is below the 95% red line. However, all the peaks in the time-integrated wavelet spectrum for Fig 1c are below the red line. Can this be clarified – is it that the integrated value is not significant because the peaks are only significant until 2050

Lines 116-7: I am not sure what this last sentence means – could it be clarified please.

Line 128: Could the sentence starting 'The disparities…' be tightened up a bit – I see what it means, but it sounds rather vague as written.

Section 3.2: Mention here that the future period is analysed to 2050.

Section 3.3: I think this analysis is just done for the pre-industrial period? It would be good to make this clear – maybe even in the section title.
The analysis in this section is good, but I found the text confusing in places. Were both the thermodynamic and dynamic aspects investigated for each of the first and second modes for example?

Lines 185-186: The analysis of spatial variability presented in sect 3.2 only covers the period to 2050 – perhaps a statement can be added there about the behaviour past-2050.

Conclusions: the first paragraphs provide a good summary of the work, but the last couple of paragraphs could be stronger. In the final paragraph do you mean the location of these devices? It would be good if this could be more explicit.

**Technical corrections**

Line 26-27 "They enlightened…"
I did not understand the first part of this sentence (relating to the historical and pre-industrial climates) – is the point that the variability for these periods is the same? It would be good if this could be re-worded to be clearer.

Line 41: It would be good to mention somewhere that this is a CMIP5 model.

Line 70: I would suggest mentioning here (with the Torrence and Compo ref) that the Morland wavelet is used, rather than in the caption for Fig 1.

Line 73: It not mentioned here which 200-year period of the 1700-year control simulation is used, although it is clear from Fig 1, maybe mention that it is the 200 years preceding the historical integration.

Line 81: I don't think the reason for analysing the shorter period is explicitly mentioned in sect 3.1. Perhaps it can be mentioned in sect 3.2.

Line 106: Sentence starting 'Those peaks and bands…' not needed  - Fig 1b could instead be referenced in the first sentence of the paragraph.

Line 125: Ref Fig2g in this sentence.

Line 138: Sentence starting 'As the first mode….', rephrase to emphasise where there is and is not agreement in the behaviour of the first and second modes.

Line 140: I don't think this sentence is needed (We looked at…)

---

## Referee Comment (RC2) · Anonymous Referee #2 · 23 May 2020

Review of "Brief communication: Arctic sea ice thickness internal variability and its changes under historical and anthropogenic forcing" by Van Achter et al.

This brief communication looks at the temporal and spatial variability of Arctic sea ice volume and thickness (respectively) using the CESM large ensemble over three multi-decadal to multi-centennial periods (pre-industrial, historical, and future). A wavelet analysis was used to explore the peak modes of variability in SIV whereas the authors used an EOF analysis to explore the spatial modes of variability in each period. The key findings of the study are that there are two peak modes of SIV variability in the pre-industrial and historical time periods centered on 8- and 16-year periods. The first of these temporal modes is shown to be related to the first spatial mode of SIT variability, which shows a strong AO signature. The relationship between the first mode of SIT and the temporal mode of SIV is made more certain by a wavelet analysis performed on the first principal component of SIT, which is found to also have a peak at 8 years. The other key finding is that both temporal modes basically vanish after 2050 when SIV reduces by 50% in winter.

The results of the analysis are original in that the temporal and spatial analyses are brought together for the first time in this way. Overall, I rate the study in terms of originality, scientific quality, significance, and presentation as fair to good.

General comments:
- The EOF analysis is informative, but have the authors considered identifying spatial modes of variability by correlating the wavelet time series associated with the peak periods for SIV with sea ice thickness? It could be interesting to see if the SIT spatial patterns agree with the EOFs, especially since the EOFs are constrained by orthogonality.

Specific comments:

L20-25:
-Labe et al., 2018 did an EOF analysis comparison with PIOMAS and CESM LE monthly ice thickness that should be referenced here.

Labe, Z., G. Magnusdottir, and H. Stern, 2018: Variability of Arctic Sea Ice Thickness Using PIOMAS and the CESM Large Ensemble. *J. Climate,* **31**, 3233–3247, https://doi.org/10.1175/JCLI-D-17-0436.1

-The Singarayer and Bamber study didn't look at sea ice thickness, just concentration.
-historical - should state that these studies all used model-based thickness reconstructions.
- the EOF mode variance numbers cited were only from the Lindsay and Zhang study, so should state this. Also could note that the spatial structures of these modes and the amount of variance they explain can be sensitive to whether (and how) the SIT time series are detrended prior to

performing the EOF or K-cluster analyses, the season and the time period considered, and the model.

L60-654:
- It needs to be clarified whether the analysis is being performed on a single ensemble member from the large ensemble or the full ensemble. If the former was done, then I would suggest at least commenting on how robust the analysis is if performed on other ensemble members. Ideally though, the full ensemble would be used. For instance, the wavelet analysis could be performed on each ensemble member and then the results of the wavelet analysis could be averaged together. It's not immediately clear to me how one would do this for EOF analysis, perhaps by appending the ensemble members to one another.
- One of the advantages of the large ensemble is that the externally-forced signal can be removed from the ensemble by subtracting the ensemble mean from each ensemble member. This makes detrending by fitting to a polynomial unnecessary and actually inferior since it requires an assumption about the functional form of the response to the forced signal.

L70:
-Should offer some more details on how the wavelet analysis was performed [software used, wavelet function (ah I now see it says in Fig. 1 caption, but should say it here), and whether any normalization was used.]

L100:
-There is still overlap though between the occurence of each peak so it's maybe not very accurate to say they don't occur at the same time.

L110:
-It might be the contouring but I'm having a hard time seeing a significant area at the 5-year period during the 2010-2025 period.

Figure 1:
-I recommend changing the units on the vertical axis from months to years since these are the units used in the text.

-What is the yellow contour representing? Presumably it's statistical significance, but if it is then I think it would be easier to see if it were colored red as in the time integrated plots. It should also be stated.

-What are the areas of white on the contour plot representing?

L120:

-Do the authors have any idea why the leading EOF mode over the historical period explains much less variance than that found in Labe et al., even though the spatial pattern looks the same? Could it be that it's due to the use of all ensemble members in Labe et al.?

-Relatedly, I'm surprised that the variance explained by the first three modes doesn't sum to a higher number. For instance, looking at Lindsay and Zhang study, the first three modes identified in their study sum closer to 60-70%.

L140:

-Why was the first mode of SIT variability only compared with ice velocity and the second mode only compared with temperature? Why not compare each mode with both ice velocity and temperature?

L155:

-I'm not sure it's accurate to say that the air temperatures are causing the variability in SIT from this analysis. They appear to be associated with each other, but this doesn't imply causation. For instance, a thermodynamic response from the ice thickness variability is just as plausible. Nonetheless it is at least consistent with Olonscheck et al. (as stated though, they looked at ice area not thickness).

Technical comments:
L85:

-I would suggest separating this sentence into two; it's currently a run-on (the two uses of the word "by")

---

## Author Comment (AC1) · 24 Jun 2020

Dear Referee, Thank you for the time that you have spent on our manuscript. We are happy with your positive response and grateful for your comments and suggestions. These certainly contributed to improving the quality of our manuscript.

Below you will find a summary of the changes that we have made throughout the manuscript to address all your suggestions. The replies to your comments are written in blue, while your comments are reproduced in black. Please, notice that line, page, and figure numbers mentioned in our rebuttal letter refer to the new version of the manuscript.

Yours sincerely and on behalf of all the co-authors,

Guillian Van Achter

**Anonymous Referee #1**

**GENERAL OVERVIEW**

This brief communication analyses the variability of Arctic sea ice thickness in pre-industrial, historical and future climate simulations from the CESM1(CAM5) coupled model. Both temporal analysis of the timeseries of sea ice volume, and spatial analysis of the sea ice thickness are presented, and results from the two analyses are brought together in the discussion. The main findings are that this model shows two peaks of temporal variability (8 and 16 years) in the pre-industrial simulation, which persist in the historical simulation, and until the middle of the 21 st century. The first mode of spatial variability is a dynamic mode related to the AO, and corresponds to the 8 year peak in temporal variability. Both the spatial and temporal variability change significantly from the 2050s when the summer sea ice is lost.

In terms of the originality, scientific quality, significance and presentation quality I asses this communication as good. The application of wavelet analysis to the SIV timeseries is interesting, and it is really good to see the temporal and spatial analysis brought together in the same piece of work.

Overall, I feel the paper could benefit from

- More clarity in the details of the analysis, and some of the explanations.
- Some improvements to figure 1
- A better bringing together of thoughts at the end of the conclusions.

Again, we thank the referee for her/his time and the detailed revision of our manuscript. We appreciated very much her/his comments, which were all taken into account in the revised version of the paper. Below, we answer point-by-point all specific comments.

**SPECIFIC COMMENTS**

Lines 46-50: While the model has been well validated for the mean state of ice thickness and extent, and for the declining in ice extent, it is of course not possible to validate the variability in the ice thickness/volume, and so I think the statement that it can be assumed that the modelled time series is an adequate proxy is perhaps too strong - the assumption is a caveat of the work.

We agree with your comment. We reformulated this statement in the new manuscript version "While it is not possible to validate the data in terms of SIT and SIV variability due to a lack of continuous observational data, the model was well validated in terms of mean state of ice thickness and extent as well as regarding the recent trends in the latter." [pg. 2, 1, 49-52].

Lines 53-54: Was the analysis done using just one of the historical and future climate ensemble members? I assume so, but it would be good to clarify this. As an aside, it would be interesting to know how robust this analysis is if it is applied to different members of the ensemble.

Since there is only member that spans from 1850 to 2005, we had decided to use only one member of the historical and future climate periods for the analysis. Thanks for your suggestion, we have now performed the multi-ensemble analysis for the study of both the temporal (wavelet) and spatial (EOF) variabilities. Please, notice that your comment agrees with one of the comments from the 2nd Referee, which also flagged the possible lack of robustness of using one member only.

We tested the robustness of our one-member EOF and wavelet analysis compared to the 30 other ensemble members. Since only one historical member is spanning the 1850-2005 time period, the historical period is now 1920-2005. For this analysis, we removed the ensemble mean from each member to obtain detrended SIT anomalies. In order to apply the EOF to the 30 members, the members were appended together over time (this method has been suggested by reviewer #2 and has already been used in the literature (Labe *et al.*, 2018)).

Figure 1 presents the first three modes of SIT variability over the historical (1920-2005) period. The modes are similar to the one of the study for the historical period (1850-2005). Figure 2 presents the first three modes of SIT variability over the future (2005-2050) time period. The first mode is similar, the second has the same pattern with small differences and the third has a different pattern of variability.

We conclude that our results for the EOF analysis over one-member are robust with the other ensemble members. The first and second modes that were described in the previous version of the manuscript are still present in the historical analysis over 30 members and the first one is still present in the future analysis. In the new manuscript, the EOF analyses for historical and future periods have been changed from a one-member to 30-member analysis.

Figure 1: Modes of Arctic SIT spatial variability. First (a), second (b) and third (c) EOF of Arctic SIT over the historical period (1920-2005). EOFs are performed over 30 ensemble members by appending them over time before applying the EOF analysis.

---

## Author Comment (AC2) · 24 Jun 2020

Dear Referee, Thank you for the time that you have spent on our manuscript. We are happy with your positive response and grateful for your comments and suggestions. These certainly contributed to improving the quality of our manuscript.

Below you will find a summary of the changes that we have made throughout the manuscript to address all your suggestions. The replies to your comments are written in blue, while your comments are reproduced in black. Please, notice that line, page, and figure numbers mentioned in our rebuttal letter refer to the new version of the manuscript.

Yours sincerely and on behalf of all the co-authors,

Guillian Van Achter

**Anonymous Referee #2**

**GENERAL OVERVIEW**

This brief communication looks at the temporal and spatial variability of Arctic sea ice volume and thickness (respectively) using the CESM large ensemble over three multi-decadal to multi-centennial periods (pre-industrial, historical, and future). A wavelet analysis was used to explore the peak modes of variability in SIV whereas the authors used an EOF analysis to explore the spatial modes of variability in each period. The key findings of the study are that there are two peak modes of SIV variability in the pre-industrial and historical time periods centered on 8- and 16-year periods. The first of these temporal modes is shown to be related to the first spatial mode of SIT variability, which shows a strong AO signature. The relationship between the first mode of SIT and the temporal mode of SIV is made more certain by a wavelet analysis performed on the first principal component of SIT, which is found to also have a peak at 8 years. The other key finding is that both temporal modes basically vanish after 2050 when SIV reduces by 50% in winter. The results of the analysis are original in that the temporal and spatial analyses are brought together for the first time in this way. Overall, I rate the study in terms of originality, scientific quality, significance, and presentation as fair to good.

Again, we thank the referee for her/his time and the detailed revision of our manuscript. We appreciated very much her/his comments, which were all taken into account in the revised version of the paper. Below, we answer point-by-point all specific comments.

The EOF analysis is informative, but have the authors considered identifying spatial modes of variability by correlating the wavelet time series associated with the peak periods for SIV with sea ice thickness? It could be interesting to see if the SIT spatial patterns agree with the EOFs, especially since the EOFs are constrained by orthogonality.

Very interesting suggestion, we thank the reviewer for that. We computed the wavelet time series associated with the first peak of variability over the pre-industrial period. Then we correlated this time series with the spatial sea ice thickness anomalies. Figure 1a (in this document) shows the correlation coefficients with 95% of significance. The correlation is quite low, but still significant (0-0.3) and the Atlantic/Pacific dipole can be found, even if the spatial pattern has some differences with the first EOF. The Pacific part is centered in the center of the Arctic Basin and the Atlantic part is smaller, restricted to the region near the North and East coast of Greenland. Furthermore, looking at Figure 1a, we note that the highest correlation values are located in areas where sea ice drift differs the most between high and low indices of the first EOF (see Figure 3 of the manuscript). This would suggest that we are indeed showing spatial variability of the 8-yr peak that is linked to the Arctic Oscillation.

On the other hand, this method may not be the most appropriate for analysing the spatial variability. Indeed, the wavelet time series associated with the first peak captures the main part of the sea ice volume anomaly. Therefore, the wavelet associated with the first peak can be seen as an approximation of the SIV anomaly. By correlating an approximation of the sea ice volume anomaly with the sea ice thickness anomaly fields, we are not surprised to obtain a high correlation near the center of the Arctic Basin, as it is the case for the correlation map between SIV and SIT (Figure 1b).

In conclusion, the spatial pattern from the wavelet time series associated with the first peak correlation with the SIT fields does have similarities with the EOFs. But from our perspective, these spatial patterns are not comparable.

Figure 1: Correlation map between the SIT anomaly (pre-industrial period) and the wavelet time series associated with the first peak of the wavelet analysis (a). Correlation map between the SIT anomaly (pre-industrial period) and the SIV anomaly (b).

**SPECIFIC COMMENTS**

L20-25:

• Labe et al., 2018 did an EOF analysis comparison with PIOMAS and CESM LE monthly ice thickness that should be referenced here

Indeed, thank you for pointing this missing reference. The paper has been added in the introduction [pg. 1, l. 21].

- The Singarayer and Bamber study didn't look at sea ice thickness, just concentration. Indeed, we removed it from the text.
- should state that these studies all used model-based thickness reconstructions. This is indeed an important information, it has been added in the text [pg. 1, l. 22].
- the EOF mode variance numbers cited were only from the Lindsay and Zhang study, so should state this. Also could note that the spatial structures of these modes and the amount of variance they explain can be sensitive to whether (and how) the SIT time series are detrended prior to performing the EOF or K-cluster analyses, the season and the time period considered, and the model.

The paragraph has been rewritten in order to take all previous comments in account [pg. 1-2, l. 20-31].

**L60-65:**

• It needs to be clarified whether the analysis is being performed on a single ensemble member from the large ensemble or the full ensemble. If the former was done, then I would suggest at least commenting on how robust the analysis is if performed on other ensemble members. Ideally though, the full ensemble would be used. For instance, the wavelet analysis could be performed on each ensemble member and then the results of the wavelet analysis could be averaged together. It's not immediately clear to me how one would do this for EOF analysis, perhaps by appending the ensemble members to one another.

We thank the reviewer for this comment. Since only one member spans from 1850 to 2005 we had decided to use only one historical member for the analysis.

We tested the robustness of our one-member EOF and wavelet analysis compared to the 30 other ensemble members. Since only one historical member is spanning the 1850-2005 time period, the historical period is now 1920-2005. For this analysis, we removed the ensemble mean from each member to obtain detrended SIT anomalies. In order to apply the EOF to the 30 members, the members were appended together over time.

Figure 2 presents the first three modes of SIT variability over historical (1920-2005) period. The modes are similar to the one of the study for the historical period (1850-2005). Figure 3 presents the first three modes of SIT variability over future (2005-2050) time period. The first mode is similar, the second has the same pattern with small differences and the third has a different pattern of variability.

We conclude that our results for the EOF analysis over one-member are robust with the other ensemble members. The first and second mode that were described in the previous version of the manuscript are still present in the historical analysis over 30 members and the first one is still present in the future analysis. In the new manuscript, the EOF analyses for historical and future period has been changed from a one-member to a 30-member analysis.

---

## Author Response (AR1)

Dear Dr. Michel Tsamados,

Here are the revised manuscript and the answers to your comments.

-Mean state of sea ice thickness. Be more specific on how this validation was performed, i.e. satellite data or PIOMAS.

The validation of the Arctic sea ice extent and thickness was done using satellite observation (Jahn et al., 2016;Swart et al., 2015;Barnhart et al., 2016) and also using PIOMAS (Labe et al., 2018).

- Appending procedure seems odd as one would expect a unnatural periodicity (i.e. the duration of each ensemble member run). Was this done for the historical (1920-2005) and future scenarios? Clarify the method a little bit more.

This was done for pre-industrial, historical (1920-2005) and future (2006-2050) periods. Since the sea ice thickness fields are preprocessed by removing the ensemble mean, we do not expect to see a periodicity link to the appending procedure. The appending procedure for analysing the spatial variability for all ensemble member was proposed by Referee 2 and allready used in previous works (Labe et al., 2018). The method consists of removing the ensemble mean to each member, append the SIT fields over time all together, and then apply the PCA. The method is described in the updated paper [p. 3, l. 65-71].

-Sign on new figure 1 different. Is that an error?

This is not an error, the sign of a PCA mode does not have meaning.

- Add some comments about how you expect your result to fare with other CMIP models.

This will depend on how the CESM-LE differs from the CMIP models in terms of sea ice state. The CESM-LE sea ice mean state is well validated. Both CESM-LE and CMIP5 sea ice trends over the observational period are nearly identical (Barnhart et al., 2016). So, we expect that the main mode of Arctic sea ice spatial and temporal variability of the CMIP simulated sea ice would look similar to our results.

[revised manuscript text omitted]